# Influence of Temperature, Solvent and pH on the Selective Extraction of Phenolic Compounds from Tiger Nuts by-Products: Triple-TOF-LC-MS-MS Characterization

**DOI:** 10.3390/molecules24040797

**Published:** 2019-02-22

**Authors:** Elena Roselló-Soto, Francisco J. Martí-Quijal, Antonio Cilla, Paulo E. S. Munekata, Jose M. Lorenzo, Fabienne Remize, Francisco J. Barba

**Affiliations:** 1Nutrition and Food Science Area, Preventive Medicine and Public Health, Food Science, Toxicology and Forensic Medicine Department, Universitat de València, Faculty of Pharmacy, Avda; Vicent Andrés Estellés, s/n, Burjassot, 46100 València, Spain; eroso2@alumni.uv.es (E.R.-S.); francisco.j.marti@uv.es (F.J.M.-Q.); antonio.cilla@uv.es (A.C.); 2Centro Tecnológico de la Carne de Galicia, Avda. Galicia No. 4, Parque Tecnológico de Galicia, San Cibrao das Viñas, 32900 Ourense, Spain; pmunekata@gmail.com (P.E.S.M.); jmlorenzo@ceteca.net (J.M.L.); 3QualiSud, Université de La Réunion, CIRAD, Université Montpellier, Montpellier SupAgro, Université d’Avignon, 2 rue J. Wetzell, F-97490 Sainte Clotilde, France; fabienne.remize@univ-reunion.fr

**Keywords:** polyphenols, tiger nut, by-products, solvent extraction, horchata de chufa, triple TOF-LC-MS-MS

## Abstract

The aim of this study was to assess the effect of temperature, solvent (hydroethanolic mixtures) and pH on the recovery of individual phenolic compounds from “horchata” by-products. These parameters were optimized by response surface methodology and triple-TOF-LC-MS-MS was selected as the analytical tool to identify and quantify the individual compounds. The optimum extraction conditions were 50% ethanol, 35 °C and pH 2.5, which resulted in values of 222.6 mg gallic acid equivalents (GAE)/100 g dry matter and 1948.1 µM trolox equivalent (TE)/g of dry matter for total phenolic content (TPC) and trolox equivalent antioxidant capacity (TEAC), respectively. The extraction of phenolic compounds by the conventional solvent method with agitation was influenced by temperature (*p* = 0.0073), and more strongly, by the content of ethanol in the extraction solution (*p* = 0.0007) while the pH did not show a great impact (*p* = 0.7961). On the other hand, the extraction of phenolic acids was affected by temperature (*p* = 0.0003) and by ethanol amount (*p* < 0.0001) but not by the pH values (*p* = 0.53). In addition, the percentage of ethanol influenced notably the extraction of both 4-vinylphenol (*p* = 0.0002) and the hydroxycinnamic acids (*p* = 0.0039). Finally, the main individual phenolic extracted with hydroethanolic mixtures was 4-vinylphenol (303.3 μg/kg DW) followed by spinacetin3-*O*-glucosyl-(1→6)-glucoside (86.2 μg/kg DW) and sinensetin (77.8 μg/kg DW).

## 1. Introduction

“Horchata de chufa” is a typical beverage from the Valencian Community. It is obtained from tiger nuts (*Cyperus esculentus*), which are tuberous rhizomes that protrude from the tips of the plant’s roots under the ground [1]. During “horchata” preparation a large amount of waste and by-products are obtained, representing “horchata” by-products ~60% of the total amount of the raw material used to obtain the beverage [2]. These by-products are a source of polysaccharides, fiber, oil (rich in oleic acid) and antioxidant compounds (e.g., vitamin E and polyphenols), among others [3].

Some previous studies have evaluated the potential application of “horchata” by-products for the preparation of new meat products, due to their high content in fiber [4,5,6]. However, the exploitation of “horchata” by-products as a source of phenolic compounds for food industries, nutraceuticals and cosmetics has not been widely explored. Some existing studies have evaluated the impact of the use of enzyme pre-treatments alone or combined with high-pressure to extract phenolic compounds from tiger nuts [7]. In this line, a previous study conducted by our research group evaluated the impact of conventional solvent extraction using a combination of binary mixtures consisting of ethanol and water at different percentages, at different temperatures and extraction times to recover total phenolic compounds and total flavonoids with antioxidant capacity from “horchata” by-products, obtaining promising results, particularly with the use of mild heating (up to 60 °C) and hydroethanolic solvents (0%–50% ethanol) [8]. Another relevant aspect of phenolic compound extraction is the selection of an appropriate pH that can influence the yield and stability of phenolic compounds. Acidic conditions are associated with higher extraction yields (higher interaction of phenolic compounds with the solvent) on different vegetable sources of phenolic compounds [9,10].

However, only spectrophotometric methods were used, as it was a preliminary study. It is well known that it is not only important to evaluate the total amount of polyphenols but also to characterize their profile as the biological activity differs according to the targeted compound. Accordingly, the use of chromatographic techniques is currently encouraged to establish the structure and activity of bioactive compounds (that can be complemented by less specific but informative spectrophotometric methodologies) to estimate the impact of conventional and non-conventional extraction processes, processing and bioaccessibility outcomes [11]. Therefore, in the present study, the impact of temperature, solvent (hydroethanolic mixtures) and pH on the recovery of individual phenolic compounds from “horchata” by-products was evaluated. For this purpose, a response surface methodology (RSM) approach was used to optimize the extraction. Moreover, triple-TOF-LC-MS-MS was selected as the analytical tool to identify and quantify the individual compounds. In addition, the results were compared to those obtained after using supercritical carbon dioxide and Folch extraction methodology.

## 2. Results and Discussion

### 2.1. Impact of Temperature, Solvent and pH on the Selective Extraction of Total Phenolic Compounds (TPC) and Trolox Equivalent Antioxidant Capacity (TEAC) from Tiger Nuts by-Products

The conventional extraction with hydroethanolic mixtures was optimized according to a Box-Behnken design in order to maximize the TEAC and TPC values. The TPC and TEAC values for each extraction are shown in Table 1. 

The TPC and TEAC values ranged from 186.52 to 222.58 mg GAE/100 g of dry matter and 617.80 to 1948.07 µM TE/g of dry matter, respectively. The best condition according to the factorial design for TPC and TEAC was 35 °C, 50% ethanol and pH 2.5. Our TPC values were higher than those obtained by Ogunlade et al. [12] who found values of 115.70 mg GAE/100 g of tiger nut in roasted tubers. In addition, Oladele et al. [13] noticed TPC values of 351 and 134 mg/100 g for yellow and brown tiger nuts, respectively. Koubaa et al. [14] obtained TPC values of 4.53–6.21 mg GAE/100 g of oil and 4.71–5.29 GAE/100 g of oil, for supercritical fluids (SC-CO_2_) and mechanical expression (ME) extractions, respectively. Parker et al. [15] found TPC values ranging from 5.63 to 64.9 mg/100 g for tiger nut, whereas Badejo et al. [16] obtained TPC values of 21.67 mg/100 mL for a tiger nut aqueous extract drink.

Moreover, Roselló-Soto et al. [8] reported that TPC values obtained from “horchata” by-products according to the solvent used, temperature and extraction time, showing that the highest TPC values were obtained using 25% ethanol (*v*/*v*), at 60 °C with an extraction time of 3 h. The difference in the TPC values could be due to raw material studied but also to the methods of extraction and analysis.

The influence of pH (2.5–12), temperature (25–50 °C) and volume of ethanol (0%–50%) to obtain phenolic compounds by conventional extraction with an ethanol:water mixtures was analyzed using a response surface methodology (RSM). As can be seen in Figure 1A, the extraction of phenolic compounds by the conventional method was influenced by temperature (*p* = 0.0073), and more strongly, by the content of ethanol in the extraction solution (*p* = 0.0007). On the contrary, the pH did not show a great impact (*p* = 0.7961). Regarding temperature, we appreciated that at pH = 7, the optimum value in an extraction without ethanol was 37 °C. However, at the highest studied ethanol concentration (50%), the optimum temperature was increased up to 43.5 °C. Furthermore, increasing the ethanol percentage led to a clear increase in extraction yield at temperatures above 40 °C, but on the contrary, this improvement was less clear at room temperature (25 °C). As proposed elsewhere, when temperature increases, the integrity of the cell wall is altered, and therefore there is a greater contact of the cellular components, among them the polyphenols, with the extraction solution [17]. However, over a certain threshold, despite the increased extraction of polyphenols due to matrix degradation, a decrease of several of these bioactive compounds can happen due to thermal lability, as observed in Figure 1A.

These values fully agree with those obtained by Moreira et al. [18], who observed that using a concentration of 50% ethanol, a 2-fold increase in the values of TPC from apple tree wood was obtained, compared to control samples without ethanol. In addition, they also found that temperature had an important impact on the extraction, getting a greater extraction at 55 °C compared to 20 °C. Vatai et al. [19] also reached the same conclusions, obtaining the maximum TPC values in Refosk (red grape marc) and lyophilized elderberry with an optimum percentage of 50% ethanol at 60 °C. On the contrary, Roselló-Soto et al. [8] observed an 82% increase in the TPC value from tiger nuts by-products with an ethanol volume of 50% at 60 °C during 2 h of extraction, but this changed at 3 h, obtaining the maximum TPC values with an ethanol concentration of 25%.

Figure 1B shows the main effects observed for antioxidant capacity at different temperatures, ethanol concentration and pH. It is clearly observed how an increase in the ethanolic fraction during the extraction of the polyphenols significantly increased the antioxidant capacity of the samples (*p* = 0.0029). Besides, neither the extraction temperature nor the pH influenced the antioxidant capacity (*p* = 0.2328 and 0.3635, respectively). Roselló-Soto et al. [8] also obtained an increase in the antioxidant capacity (TEAC assay) of extracts from “horchata” by-products when they used an ethanol concentration of 50% and an increase in temperature up to 60 °C. In our case, there was also an increasing trend (*p* > 0.05) in the antioxidant capacity when the temperature was augmented, as well as when highly acidic and alkaline conditions were used. However, as it was mentioned above, these increases are not significant. Moreover, Li et al. [20] also indicated that the TEAC values of extracts obtained from *Gordonia axillaris* increased with an ethanol volume of 40% and a temperature of 40 °C, but decreased when these values were higher. According to the authors, this could be due to the degradation of some thermolabile antioxidant compounds. Moreira et al. [18] studied the extraction of polyphenols and antioxidant capacity from apple tree wood, observing the highest antioxidant capacity (measured using FRAP assay) after using a temperature of 55 °C and 50% ethanol volume. Similarly, Rusu et al. [21] indicated that increasing ethanol percentage in the solvent (from 50% to 95%) and extraction temperature (from 20 to 40 °C) were associated with higher TPC and TEAC values in walnut septum extract. Interestingly, the authors also obtained higher TPC and TEAC values by carrying out the extraction with 50% ethanol solution at 20 °C. In addition, Bamba et al. [22] obtained an increase in the antioxidant capacity (measured using DPPH assay) with augmented temperature, but since the content of phenolic compounds decreased, this increase could be due to the presence of other antioxidant compounds.

The ANOVA results (Table 2) show that not only these parameters by themselves can affect the extraction yield, but the combination of them also can produce significant changes. This is the case of TPC, in which the combination of temperature and ethanol can modify TPC extraction (*p* = 0.0318).

### 2.2. Impact of Temperature, Solvent and pH on the Selective Extraction of Individual Phenolic Compounds from Tiger Nuts by-Products

The profile and content of specific phenolic compounds extracted with ethanol:water mixtures from tiger nut by-products using the RSM methodology for optimization and after analyzing the extracts by Triple-TOF-LC-MS-MS are shown in Table 3 and Figure 2.

Specifically, as can be seen in Figure 2A, both pH and temperature had a great influence on the extraction of lignans (*p* = 0.0256 and 0.0251, respectively). The maximum yield was observed at pH 10.62 and 43.3 °C. However, at room temperature (25 °C) the optimum pH dropped to 6.2. This outcome agrees with the data previously obtained by Tu et al. [23] who found an optimum pH in the range 5.5–6 for the extraction of lignans at room temperature for *Fructus forsythiae*.

Phenolic acid extraction was also influenced in a very large amount by temperature (*p* = 0.0003) and by the volume of ethanol (*p* < 0.0001) but not by the pH (*p* = 0.53) (Figure 2B). In this case, it should be noted how the maximum extraction yield was obtained for a temperature of 50 °C and an ethanol content of 41.4%. Elez-Garofulić et al. [24] also observed an increase in the extraction of phenolic acids by increasing the temperature using the microwave-assisted extraction on sour cherry Marasca, obtaining an optimum temperature of 70 °C. A similar trend was also reported by Waszkowiak et al. [25], who studied the influence of the percentage of ethanol used for the extraction of phenolic compounds of flaxseeds extracts, ranging from 60% and 90%, and they observed that the best ratio for phenolic acids corresponded to 60% ethanol in water.

Furthermore, the percentage of ethanol influenced notably the extraction of both 4-vinylphenol (*p* = 0.0002) and the hydroxycinnamic acids (*p* = 0.0039) (Figure 2C,D). This finding can be explained by the polarity of the solvent, since increasing the volume of ethanol in water increases the polarity. As it is known, these compounds are polar, so a more polar solvent will extract them better. Other authors such as Woźniak et al. [26], Chew et al. [27] and Paini et al. [28] also used mixtures with ethanol to improve the yield of polyphenols extraction. In the case of flavonoids, no statistically significant differences were found on the extraction yield of dihydroxybenzoic acids and flavones when varying these parameters.

In Table 4 the ANOVA results are shown for the influence of temperature, ethanol and pH in the extraction of the compounds explained in Figure 2. As can be seen, for the lignans extraction the combination of temperature and pH had a significant effect (*p* = 0.0344). This could be explained by the increase of solubility, which improved the extraction of these compounds.

Possible beneficial effects of polyphenols on human health are the subject of increasing scientific interest. For example, phenolic acids and lignans have been shown to have a positive hepatoprotective action [29]. The anti-inflammatory action of ferulaldehyde was also found in mice by other authors [30]. For all these reasons, it is necessary to consider the most appropriate extraction conditions depending on the compounds desired to obtain.

### 2.3. Optimization and Validation of the Extraction Conditions

The combination of critical parameters (temperature, ethanol and pH), which allowed to obtain the highest TPC yield and TEAC value. To do this, an optimization based on desirability was used. Theoretically, in the case of TPC, optimum values of 229.29 mg GAE/100 g of dry matter were obtained with conditions of 43.7 °C, 50% ethanol and pH = 2.5. For the antioxidant capacity, the optimum value obtained was 1846.34 µM TE/g of dry matter at a temperature of 50 °C, a volume of ethanol of 50% and a pH = 2.5. As can be seen in Table 1, the maximum extraction of TPC and the maximum TEAC values were obtained experimentally with the conditions of 35 °C, 50% ethanol and pH = 2.5 (222.58 ± 2.16 mg GAE/100 g of dry matter and 1948.07 ± 434.18 µM TE/g of dry matter respectively). These results are close to those expected theoretically, so we can affirm that the method has been validated for TPC and TEAC.

### 2.4. Comparison of Hydroethanolic Extraction of Individual Phenolics Compounds from Horchata by-Products with Those of Folchand Supercritical-CO_2_ Extraction

The data on individual phenolics obtained in the present study were compared to those obtained after conventional extraction method (Folch) and an innovative extraction method (supercritical CO_2_ extraction). The results for the comparison were obtained from a previously published article about the extraction of phenolic compounds in the oil fraction of “horchata” by-products [31].

It can be noticed that there was a great difference in the compounds obtained with the three different extraction techniques. As for conventional extraction with Folch method, the major compound obtained by far was 4-vinylphenol (216.9 μg/kg DW), followed by p-coumaric acid (25.35 μg/kg DW) and benzoic acid (13.54 μg/kg DW). In contrast, the main compounds extracted by SC-CO_2_, especially at 40 MPa, were isohydroxymatairesinol (399.44 μg/kg DW), scopoletin (93.24 μg/kg DW) and caffeic acid (30.66 μg/kg DW). In the present study (see Table 2), the main individual phenolic extracted with hydroethanolic mixtures was 4-vinylphenol (303.3 μg/kg DW) followed by spinacetin3-O-glucosyl-(1→6)-glucoside (86.2 μg/kg DW) and sinensetin (77.8 μg/kg DW).

It is worth noting that 4-vinylphenol was the predominant phenolic compound obtained, after using both conventional methods, despite the different polarity of solvents employed, though higher extraction is accomplished with hydroethanolic mixtures. Moreover, a great difference was also observed between the compounds obtained after conventional extraction (Folch and hydroethanolic mixtures) and those obtained by SC-CO_2_. As already mentioned, a more polar solvent facilitates the extraction of more phenolic compounds, therefore the low yield obtained in the SC-CO_2_ extraction is not surprising. We should also keep in mind that the way to prepare the sample is different. In the case of conventional extraction, homogenization is carried out at 10,000× rpm, while for SC-CO_2_ extraction the sample is only milled. This protocol differences greatly modify the accessibility of the solvents to the intracellular compounds, since in the case of conventional extraction the samples were homogenized with the solvent, breaking the cellular structures, whereas in the SC-CO_2_ extraction only the particle size of the samples was reduced.

These findings agree with the data previously reported by Parker et al. [15] who demonstrated that tiger nuts skins are richer in *p*-coumaric acid than tiger nuts tubers, being this compound the fourth most important (48 μg/kg DW) in “horchata” by-products after hydroethanolic extraction. In addition, Ezeh et al. [7] studied the polyphenols present in tiger nuts, finding mainly *trans*-cinnamic acid. Oladele et al. [13] also determined the phenolic profile of tiger nuts, considering the yellow and brown varieties. They obtained differences between both species, since in yellow tiger nuts the main compounds were ferulic acid (~58 mg/100 g), *p*-hydroxybenzoic acid (~29 mg/100 g), *p*-hydroxybenzaldehyde (~16 mg/100 g) and vanillic acid (~6 mg/100 g), whereas in brown tiger nut, vanillic (~15 mg/100 g), *p*-coumaric (~17 mg/100 g), caffeic (~15 mg/100 g), ferulic (~34 mg/100 g) and sinapinic acids (~21 mg/100 g) were predominant.

## 3. Material and Methods

### 3.1. Chemicals and Reagents

Sodium hydroxide (NaOH), sodium carbonate (Na_2_CO_3_), and acetone were obtained from J.T.Baker (Deventer, Holland). ABTS radical (2,2′-azino-bis(3-ethylbenzothiazoline-6-sulphonic acid), Trolox (6-hydroxy-2,5,7,8-tetramethylchroman-2-carboxylic acid), Folin-Ciocalteau phenol reagent 1N, gallic acid, sodium nitrite (NaNO_2_), formic acid (HPLC grade), potassium persulfate (K_2_S_2_O_8_) and ethanol p.a. (99.5%) were purchased from Sigma-Aldrich (Steinheim, Germany). Sodium hydrogen carbonate (reagent grade; 99.7%) and methanol (reagent grade; 99.9%) were purchased from Scharlau (Barcelona, Spain). Deionized water was obtained from Millipore (Bedfore, MA, USA).

### 3.2. Samples

A conventional process was used for obtaining “horchata” from tiger nuts with a denomination of origin “Chufa de Valencia” (*Cyperus esculentus*). Then “horchata” by-products were taken and provided by the “Consejo Regulador D.O. Chufa de Valencia” (Valencia, Spain). Afterwards, they were dried at 60 °C for 72 h using a Memmert UFP 600 air-circulating oven (Schwabach, Germany) and ground for obtaining uniform particle size. Finally, they were vacuum packed until needed.

### 3.3. Extraction at Different Temperatures, Ethanol:Water Mixtures and pH

The conditions for solid–liquid extraction were selected based on a previous study [32]. First, one gram of dehydrated “horchata” by-product was weighed and fifteen milliliter of the hydroethanolic mixtures at different ethanol concentrations (0%, 25% and 50%, *v*/*v*) and different pH (2.5, 7.25 and 12), adjusted with NaOH or HCl was used for the extraction. The beakers with the samples were then placed and stirred on a plate with a magnetic stirrer. To avoid the evaporation of the solvent during the extraction, the samples were covered with aluminum foil. The temperature was adjusted to 25, 35 and 50 °C in each of the stirring plate’s rows. The extraction was carried out for 3 h. The obtained samples were filtered through Whatman No. 1 and used for the determination of phenolic compounds.

### 3.4. Total Antioxidant Capacity

TEAC (Trolox equivalent antioxidant capacity) assay was used for the determination of the total antioxidant capacity [8]. Twenty-five milliliter of ABTS (7 mM) was mixed with 440 μL of K_2_S_2_O_8_ (140 mM) and kept at room temperature for 12–16 h under darkness. For the determination, the absorbance of ABTS^•+^ working solution was measured at a wavelength of 734 nm on a Perkin-Elmer UV/Vis Lambda 2 spectrophotometer (Perkin-Elmer, Jügesheim, Germany) to obtain the initial absorbance (A_0_). When the absorbance of the mixture was 0.700 ± 0.020, 100 µL of the extracts appropriately diluted were added, and the absorbance was measured at 20 min (A_f_). The following equation was used to calculate the inhibition percentage (%) of the samples:% Inhibition = (1 − A_f_/A_0_) × 100,
where A_0_ is the absorbance at the initial time and A_f_ is the absorbance obtained after 20 min.

The results were expressed as micromolar Trolox equivalent (μM TE)/g of dry matter.

### 3.5. Determination of Total Phenolic Content (TPC)

The method previously reported [33], with some modifications [34] was used. To sum up, 500 µL of extract was mixed with 4.5 mL of distilled water and then 1 mL of the 2% Na_2_CO_3_ solution (*w*/*v*), and 0.25 mL of the Folin-Ciocalteau reagent (1N) were added. The mixture was left to stand for one hour under darkness at room temperature. Afterwards, the absorbance was measured at 765 nm. TPC was determined by interpolating the absorbance values in a calibration curve using gallic acid standard (10 μg/mL) at different concentrations between 0 and 5 μg/mL. The results were expressed as mg equivalents of gallic acid (GAE)/100 g of dry matter.

### 3.6. Triple TOF–LC–MS–MS Characterization of Phenolic Compounds

The phenolic profile characterization and quantification was performed according to the previously described method [31], using an Agilent 1260 Infinity (Agilent, Waldbronn, Germany) with a Waters UPLC C18 column 1.7 µm (2.1 × 50 mm) Acquity UPLC BEH.C18 (Waters, Cerdanyola del Vallès, Spain) for the separation of the main phenolic compounds in the samples. Moreover, a TripleTOF™ 5600 LC/MS/MS system (AB SCIEX, Foster City, CA, USA) was utilized for the identification. For that purpose, a mobile phase consisting of solvent A (water, 0.1% formic acid) and solvent B (methanol, 0.1% formic acid) was used as follows: 0 min 90% A; 13 min 100% (B); 15 min 90% A. Five microliter and 0.4 mL/min were the injection volume and flow rate, respectively.

MS data were obtained between 80 and 1200 *m*/*z* on negative mode, and the IDA acquisition method was carried out in the survey scan type (TOF-MS) using the dependent scan type (product ion). Ion spray voltage (−4500 V); declustering potential (90 V); collision energy (−50 V); temperature with 25 psi curtain gas (400 °C); 50 psi for both ion source gas 1 (GC1) and ion source gas 2 (GS2) were used as the main parameters for the MS analysis. 

The IDA MS/MS analysis was carried out with ion tolerance of 50 mDa, 25 V collision energy and activated dynamic background subtract. The software analyst PeakView1.1 (AB SCIEX, Foster City, CA, USA) and its applications (XIC Manager and Formula Finder) were used for data acquisition and processing. Finally, an external calibration curve using resveratrol as standard was prepared for the quantification of phenolic compounds.

### 3.7. Experimental Design and Statistical Analyses

Box-Behnken design with three levels (maximum, minimum, and central) of each independent variable, temperature (25–50 °C), the concentration of ethanol (0%–50%), and pH (2.5–12), leading to 15 combinations of these variables. Independent variable levels were selected accounting for the sample and the potential degradation of thermolabile antioxidant compounds after high temperatures (>50 °C). The combinations included temperature–ethanol–pH conditions with an intermediate level (central point) of the three variables replicated three times, which was used to check the reproducibility and stability of the results. Experiments were randomized to minimize the systematic bias in the observed responses due to extraneous factors and for higher precision. In addition, we studied whether there were correlations between a pair of variables. The significant differences (*p* < 0.05) between the results were calculated by analysis of variance (ANOVA), using the least significant difference (LSD) test to indicate the samples between which there were differences. All statistical analyses were performed using the software Statgraphics^®^ Centurion XV (Statpoint Technologies, Inc., The Plains, VA, USA). 

## 4. Conclusions

The optimization of phenolic compounds extraction by response surface methodology, followed by Triple-TOF-LC-MS-MS characterization, indicated that temperature and ethanol content are more important variables than pH. The optimum extraction conditions for total phenolic content and antioxidant activity were 50% ethanol, 35 °C and pH 2.5, which could obtain values of 222.6 mg GAE/100 g of dry matter and 1948.1 µM TE/g of dry matter for TPC and TEAC, respectively. The optimized extraction condition also positively influenced the extraction of the main individual phenolic compounds of tiger nuts by-products, particularly 4-vinylphenol and hydroxycinnamic acids. Therefore, tiger nuts by-products can be explored as a valuable source of phenolic compounds with potential applications in food industries, nutraceuticals and cosmetics.

## Figures and Tables

**Figure 1 molecules-24-00797-f001:**
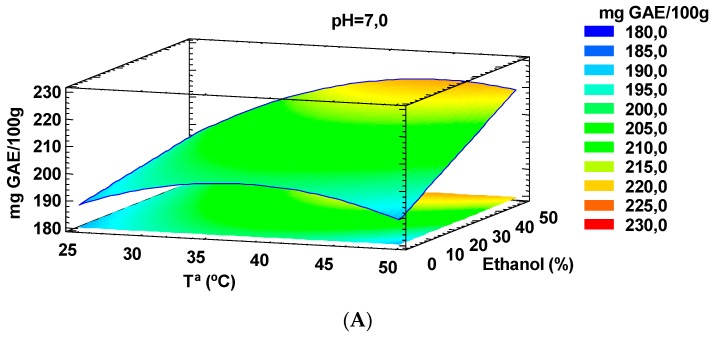
Plot for the influence of extraction condition parameters in total phenolic content (mg gallic acid equivalents ((GAE)/100 g of dry matter) (**A**) and the main effects chart for antioxidant activity (**B**) using solid-liquid extraction. One gram of tiger nuts by-products was extracted using different temperatures (25–50 °C), ethanol:water mixture (0%–50% ethanol) and pH (2.5–12).

**Figure 2 molecules-24-00797-f002:**
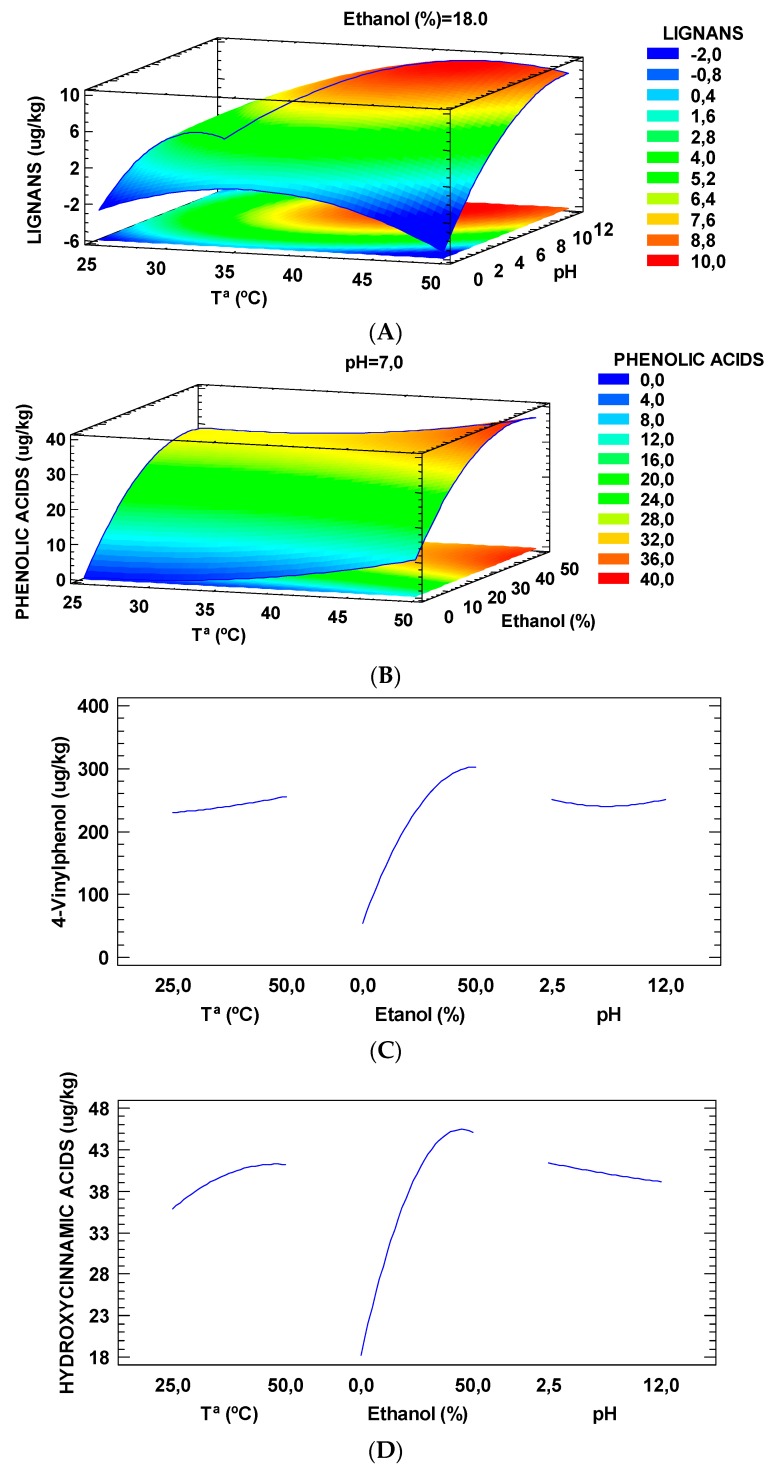
Response surface plot for the influence of extraction condition parameters in lignans (**A**) and phenolic acids (**B**). Main effects chart for 4-vinylphenol (**C**) and hydroxycinnamic acids (**D**) using solid-liquid extraction. 1 g of tiger nuts by-products was extracted by varying temperature (25–50 °C), ethanol:water mixture (0%–50%) and pH (2.5–12).

**Table 1 molecules-24-00797-t001:** Total phenolic content (TPC) and total antioxidant capacity (TEAC) from “horchata” by-products obtained after conventional extraction with hydroethanolic mixtures.

Run#	T (°C)	Ethanol (%)	pH	TPC (mg GAE/100 g of Dry Matter)	TEAC Values (µM TE/g of Dry Matter)
1	25	0	7.25	186.52 ± 4.58	705.31 ± 18.45
2	25	25	2.5	197.74 ± 1.38	996.68 ± 5.26
3	25	25	12	196.74 ± 2.09	1325.68 ± 109.73
4	25	50	7.25	200.06 ± 2.34	1185.43 ± 284.17
5	35	0	2.5	203.95 ± 2.64	617.80 ± 24.15
6	35	0	12	207.20 ± 13.06	862.10 ± 50.04
7	35	25	7.25	211.79 ± 3.75	1110.00 ± 216.41
8	35	25	7.25	206.92 ± 8.28	1094.15 ± 171.80
9	35	25	7.25	203.64 ± 2.39	1091.04 ± 50.88
10	35	50	2.5	222.58 ± 2.16	1948.07 ± 434.18
11	35	50	12	215.72 ± 1.51	1785.94 ± 84.12
12	50	0	7.25	186.78 ± 0.94	983.24 ± 101.28
13	50	25	2.5	209.17 ± 3.32	1101.18 ± 79.84
14	50	25	12	210.38 ± 4.31	1328.06 ± 76.90
15	50	50	7.25	220.48 ± 2.47	1644.27 ± 28.53

**Table 2 molecules-24-00797-t002:** Analysis of variance (ANOVA) results for each effect in response surface methodology (RSM) of total phenolic compounds (TPC) and trolox equivalent antioxidant capacity (TEAC) values.

	TPC		TEAC	
Source	*p*-Value	Sig.	*p*-Value	Sig.
A: Tª (ºC)	0.0073	**	0.2328	n.s.
B: Ethanol (%)	0.0007	***	0.0029	***
C: pH	0.7961	n.s.	0.3635	n.s.
AA	0.0036	***	0.6071	n.s.
AB	0.0318	*	0.8815	n.s.
AC	0.7439	n.s.	0.9067	n.s.
BB	0.9735	n.s.	0.5495	n.s.
BC	0.2318	n.s.	0.3976	n.s.
CC	0.0497	*	0.3016	n.s.

Sig: significance; ns: not significant, * (*p* < 0.05); ** (*p* < 0.01); *** (*p* < 0.001).

**Table 3 molecules-24-00797-t003:** Individual phenolic compounds (μg/kg) (lignans, flavones, flavonoids, dihydroxybenzoic acids, hydroxycinnamic and phenolic acids) determined by Triple-TOF-LC-MS-MS from “horchata” by-products obtained after conventional extraction with hydroethanolic mixtures.

T^a^ (°C)	Ethanol (%)	pH	1-AP	7-HS	Cyanidin	Ethyl Vanillin	4-Vinylphenol	Sinensetin	SGG	Cinnamic Acid	DPC	PC	FG+FG	Ferulaldehyde	4-HB+BA
25	0	7.25	ND	ND	ND	10.40	85.00	ND	ND	3.30	10.40	ND	4.40	ND	ND
25	25	2.5	ND	ND	ND	10.90	209.2	ND	ND	4.00	31.50	ND	ND	1.80	23.00
25	25	12	ND	ND	ND	10.60	258.8	ND	ND	ND	34.90	ND	ND	3.10	21.40
25	50	7.25	ND	ND	ND	11.00	264.3	ND	ND	ND	38.30	ND	ND	3.70	24.50
35	0	2.5	1.70	ND	ND	14.90	126.2	ND	35.50	6.10	13.50	ND	8.80	0.50	ND
35	0	12	ND	3.70	ND	21.50	20.10	ND	25.80	ND	ND	0.20	7.40	ND	0.60
35	25	7.25	4.30	3.00	2.30	10.60	229.9	50.10	85.00	ND	ND	28.00	8.70	3.00	21.30
35	25	7.25	4.40	4.10	2.40	11.60	248.6	50.00	86.00	ND	ND	33.50	8.00	3.90	22.80
35	25	7.25	4.30	4.50	2.00	12.20	234.7	50.00	81.90	ND	ND	31.90	8.50	3.80	22.60
35	50	2.5	3.90	ND	4.10	13.20	303.3	77.80	86.20	ND	ND	41.80	8.10	6.60	23.20
35	50	12	4.00	ND	5.50	11.10	297.4	76.10	95.40	ND	ND	38.90	ND	6.10	22.10
50	0	7.25	1.40	4.20	ND	26.00	22.30	ND	23.00	11.00	ND	ND	4.90	ND	11.40
50	25	2.5	ND	ND	ND	12.20	242.5	57.30	92.50	ND	ND	32.90	ND	5.30	25.00
50	25	12	4.60	4.40	3.50	11.90	302.6	60.40	92.80	ND	ND	42.10	9.30	5.20	28.80
50	50	7.25	ND	ND	5.30	18.70	352.7	ND	ND	ND	ND	48.00	ND	3.60	34.20

Lignans: 1-Acetoxypinoresinol (AP); 7-Hydroxysecoisolariciresinol (7-HS). Dihydroxybenzoic acid: Ethylvanillin. Hydroxycinnamic acids: Cinnamic acid; Dihydro-p-coumaric acid (DPC); p-coumaric acid (PC); Ferulicacid 4-*O*-glucoside + Feruloyl glucose (FG+FG). Phenolicacids: 4-Hydroxybenzaldehyde+Benzoic acid (4-HB+BA); Ferulaldehyde. Flavones: Sinensetin. Flavonoids: Cyanidin; Spinacetin 3-*O*-glucosyl-(1→6)-glucoside (SGG). ND: not detected.

**Table 4 molecules-24-00797-t004:** ANOVA results for each effect in RSM of TPC and TEAC values.

Source	Lignans		Phenolic Acids		4-Vinylphenol		Hydroxycinnamic Acids	
	*p*-Value	Sig.	*p*-Value	Sig.	*p*-Value	Sig.	*p*-Value	Sig.
A: T^a^ (°C)	0.0251	*	0.0003	***	0.3599	n.s.	0.3558	n.s.
B: Ethanol (%)	0.5893	n.s.	0.0000	***	0.0002	***	0.0039	***
C: pH	0.0256	*	0.5304	n.s.	0.9698	n.s.	0.6837	n.s.
AA	0.0050	**	0.0165	*	0.9056	n.s.	0.7121	n.s.
AB	0.1028	n.s.	0.5178	n.s.	0.0843	n.s.	0.4661	n.s.
AC	0.0344	*	0.1846	n.s.	0.6726	n.s.	0.1701	n.s.
BB	0.0153	*	0.0000	***	0.0219	*	0.0804	n.s.
BC	0.3209	n.s.	0.5820	n.s.	0.2235	n.s.	0.5392	n.s.
CC	0.0473	*	0.2462	n.s.	0.5964	n.s.	0,9739	n.s.

Sig: significance; ns: not significant, * (*p* < 0.05); ** (*p* < 0.01); *** (*p* < 0.001).

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
