# Peer review of "Influence of Temperature, Solvent and pH on the Selective Extraction of Phenolic Compounds from Tiger Nuts by-Products: Triple-TOF-LC-MS-MS Characterization"

_molecules, 2019, doi:10.3390/molecules24040797_

Round 1

Reviewer 1 Report

The study is well performed, the applied methodology is up-to-date, the discussions are pertinent and the bibliographic references are adequate.

I recommend accepting the manuscript for publication after minor revision, as follows:

-          Please report the antioxidant activity TEAC per dry weight of “horchata” in all manuscript (lines: 25, 79 (Table 1), 82, 311).

-          Line 27: Please elaborate what conventional method did you use?

-          Lines 90-92: Reference [8] is a previous study that belongs to you. Despite the fact that in the mentioned study you obtained better extraction yields for TPC, TFC, and AAT-TEAC at 60 °C after 3 hours of agitation, in this study you preferred to extract at slightly different parameters, temperature was set only up to 50 °C and the agitation time was only 2 hours. Please motivate the choosing of the parameters for this study.

-          Line 103: Because you use past tense in the sentence, I suggest changing “lead” to “led”.

-          Lines 106-109: I suggest using another word for “certain” (twice in the sentence) and reformulating the second part of the sentence, something like this: “However, over a certain threshold, despite the increase of extraction of polyphenols due to matrix degradation, a decrease of several of these bioactive compounds can happen due to thermal lability, as observed in Figure 1A.”

-          Line 120: Please use another word for “perfectly”.

-          Lines 127-136: I suggest the addition of a new citation in this paragraph, an article recently published in Molecules (same as References # 18 and 19): Rusu, M. E.; Gheldiu, A.-M.; Mocan, A.; Moldovan, C.; Popa, D.-S.; Tomuta, I.; Vlase, L. Process Optimization for Improved Phenolic Compounds Recovery from Walnut (Juglans regia L.) Septum: Phytochemical Profile and Biological Activities. Molecules. 2018, 23, 2814, doi:10.3390/molecules23112814.

-          Lines 151-153: Table 2 – Please separate “acid(s)” from the first part of the chemical name of the polyphenols. Add also the meaning of “ND”.

-          Lines 184-186: Please specify that in the reference [28], the two extraction methods were used to describe the phytochemical composition of the oils obtained from “horchata”.

-          Line 195: Please separate “hydroxycinnamic” from “acids”.

-          Line 227: I suggest erasing “the”.

-          Line 262: In the formula, please write “/” normal, not with subscript.

-          Line 309: I suggest using “important” instead of “crucial”.

Lines 311-312: I suggest reformulating the beginning of the sentence like this: “The optimized extraction conditions were also studied and described for the main individual phenolic compounds…” or like this: “The optimized extraction conditions also positively influenced the extraction of the main individual phenolic compounds…”

Author Response

The manuscript has been carefully rechecked, and appropriate changes have been made in accordance with the reviewers’ suggestions. We would like to thank the referees for their excellent comments. They have helped improving the manuscript quality significantly, and we believe that the manuscript now provides a more balanced and better account of the research. We have modified the manuscript accordingly, and the detailed corrections are listed below point by point. Changes to the manuscript related to reviewer #1 and #2 comments have been indicated with red and blue color, respectively.

Please do not hesitate to contact me again if further changes to the manuscript are required.

Reviewer #1: General comments

The study is well performed, the applied methodology is up-to-date, the discussions are pertinent and the bibliographic references are adequate.

I recommend accepting the manuscript for publication after minor revision, as follows:

- Please report the antioxidant activity TEAC per dry weight of “horchata” in all manuscript (lines: 25, 79 (Table 1), 82, 311).

RESPONSE: Thank you for the comment. The units of TEAC results have been updated from µM Trolox to µM Trolox/g of dry matter throughout the manuscript.

-Line 27: Please elaborate what conventional method did you use?

RESPONSE: Thank you for the comment. The statement “conventional method” was replaced by “conventional solvent method with agitation”.

- Lines 90-92: Reference [8] is a previous study that belongs to you. Despite the fact that in the mentioned study you obtained better extraction yields for TPC, TFC, and AAT-TEAC at 60 °C after 3 hours of agitation, in this study you preferred to extract at slightly different parameters, temperature was set only up to 50 °C and the agitation time was only 2 hours. Please motivate the choosing of the parameters for this study.

RESPONSE: Thank you very much for your comment. Taking into account that in our previous study, we only determined TPC, TFC, and AAT-TEAC, and we were not sure if the thermolabile polyphenols could be affected after using 60 ºC, we decided to decrease a little bit the temperature to avoid potential degradation of thermolabile compounds, as we are determining the phenolic profile. Regarding the extraction time, it was a mistake, we used 3h. It is now corrected in text.

-Line 103: Because you use past tense in the sentence, I suggest changing “lead” to “led”.

RESPONSE: Following the reviewer’s suggestion, “lead” was replaced by “led”.

- Lines 106-109: I suggest using another word for “certain” (twice in the sentence) and reformulating the second part of the sentence, something like this: “However, over a certain threshold, despite the increase of extraction of polyphenols due to matrix degradation, a decrease of several of these bioactive compounds can happen due to thermal lability, as observed in Figure 1A.”

RESPONSE: Thank you for the suggestion. The sentence was accordingly updated following the reviewer’s suggestion.

- Line 120: Please use another word for “perfectly”.

RESPONSE: Thank you for the comment. The word “perfectly” was replaced by “clearly”.

- Lines 127-136: I suggest the addition of a new citation in this paragraph, an article recently published in Molecules (same as References # 18 and 19): Rusu, M. E.; Gheldiu, A.-M.; Mocan, A.; Moldovan, C.; Popa, D.-S.; Tomuta, I.; Vlase, L. Process optimization for improved phenolic compounds recovery from walnut (Juglans regia L.) septum: Phytochemical profile and biological activities. Molecules. 2018, 23, 2814, doi:10.3390/molecules23112814.

RESPONSE: Thank you for the suggestion. The paragraph was updated to include about this new reference.

- Lines 151-153: Table 2 – Please separate “acid(s)” from the first part of the chemical name of the polyphenols. Add also the meaning of “ND”.

RESPONSE: Thank you for the comment. The names of phenolic acids in the foot note of old Table 2 (new table 3) were corrected. Also the meaning of ND was included.

- Lines 184-186: Please specify that in the reference [28], the two extraction methods were used to describe the phytochemical composition of the oils obtained from “horchata”.

RESPONSE: Thank you for the comment. A statement indicating that reference [28] (now reference [31]) explored the extraction of phenolic compounds in the oil of “horchata” by-products was included in the revised version of the manuscript.

- Line 195: Please separate “hydroxycinnamic” from “acids”.

RESPONSE: Thank you for the comment. A correction in this line was done.

- Line 227: I suggest erasing “the”.

RESPONSE: Thank you for the comment. A correction in this line was done.

- Line 262: In the formula, please write “/” normal, not with subscript.

RESPONSE: Thank you for the comment. A correction in this formula was done.

- Line 309: I suggest using “important” instead of “crucial”.

RESPONSE: Thank you for the suggestion. A correction in this line was done.

- Lines 311-312: I suggest reformulating the beginning of the sentence like this: “The optimized extraction conditions were also studied and described for the main individual phenolic compounds…” or like this: “The optimized extraction conditions also positively influenced the extraction of the main individual phenolic compounds…”

RESPONSE: Thank you for the suggestion. The sentence was accordingly updated following the reviewer’s suggestion.

Reviewer 2 Report

The manuscript describes the effect of T, pH and ethanol content on the extraction of phenolic compounds from tiger nuts by-products. In my opinion a major revision is required:

In the Abstract authors say “Finally, the main individual phenolic extracted with hydroethanolic mixtures was 4- vinylphenol (352.7 mg/kg DW) followed by spinacetin 3-O-glucosyl-(1->6)-glucoside (95.4 mg/kg DW) and sinensetin (76.1 mg/kg DW)”, but is not clear in at what conditions were these values obtained.

In the discussion, the use of the Box-Behnken design should be mentioned. Table 1 also should mention the order of the experiments. The reason for the study of these parameters in particular, as well as their ranges, should also be clarified.

The ANOVA results should appear. What is the significance of each effect? Effects on each response should be divided by topic. Interactions effects are not mentioned. Why?

Optimal parameters and response appear in Abstract and conclusions, but these results are not presented neither discussed in the discussion section. How this optimization was done? Which model was used? Which response was optimized? Was the model validated?

 TPC seems to be used to abbreviate both “Total Phenolic Content” and “Total Phenolic Compounds”. Please use for a single abbreviation, I suggest “Total Phenolic Content” which is widely used in the bibliography together with Folin-Ciocalteu method.

Legend Table 2: Some spaces are missing.

Some contradictory interpretation is also observed, for example:

Lines 122 and 123  “Besides, neither the extraction temperature nor the pH influenced the antioxidant capacity (P =0.2328 and 0.3635, respectively).” Later, in Lines 125-127: “ In our case, there was also observed an increasing trend (P> 0.05) in the antioxidant capacity when temperature was augmented, as well as when highly acidic and alkaline conditions were used.”

References: Issue and pages information is missing in the references 16,17,18,19 and 21.

Author Response

The manuscript has been carefully rechecked, and appropriate changes have been made in accordance with the reviewers’ suggestions. We would like to thank the referees for their excellent comments. They have helped improving the manuscript quality significantly, and we believe that the manuscript now provides a more balanced and better account of the research. We have modified the manuscript accordingly, and the detailed corrections are listed below point by point. Changes to the manuscript related to reviewer #1 and #2 comments have been indicated with red and blue color, respectively.

Please do not hesitate to contact me again if further changes to the manuscript are required.

Reviewer #2: General comments

The manuscript describes the effect of T, pH and ethanol content on the extraction of phenolic compounds from tiger nuts by-products. In my opinion a major revision is required:

In the Abstract authors say “Finally, the main individual phenolic extracted with hydroethanolic mixtures was 4- vinylphenol (352.7 mg/kg DW) followed by spinacetin 3-O-glucosyl-(1->6)-glucoside (95.4 mg/kg DW) and sinensetin (76.1 mg/kg DW)”, but is not clear in at what conditions were these values obtained.

RESPONSE: Thank you for the comment. The values indicated in both Abstracts and Result and Discussion sections were updated. Now these values reflect the optimum extraction condition (35 °C, 50% ethanol, and pH 2.5).

In the discussion, the use of the Box-Behnken design should be mentioned. Table 1 also should mention the order of the experiments. The reason for the study of these parameters in particular, as well as their ranges, should also be clarified.

RESPONSE: Thank you for the comment. A statement regarding the use of Box-Behnken design was included in the discussion and the number of experiments was indicated in Table 1. The selection of extracting variables and the range of values were also indicated in the introduction.

The ANOVA results should appear. What is the significance of each effect? Effects on each response should be divided by topic. Interactions effects are not mentioned. Why?

RESPONSE: Thank you for the suggestion. ANOVA results and the significance have been included in tables (Tables 2 and 4), and interaction effects have been mentioned:

 “ANOVA results (Table 2) show that not only these parameters can affect the extraction yield, but the combination of them also can promote significant changes. This is the case of TPC, in which the combination of temperature and ethanol can modify TPC extraction (P=0.0318).”

“In Table 4 are shown ANOVA results for the influence of temperature, ethanol and pH in the extraction of the compounds explained in Fig. 2. As can be seen, for the lignans extraction the combination of temperature and pH has a significant effect (P=0.0344). This could be explained by the increase of solubility, which improves the extraction of these compounds.”

Optimal parameters and response appear in Abstract and conclusions, but these results are not presented neither discussed in the discussion section. How this optimization was done? Which model was used? Which response was optimized? Was the model validated?

RESPONSE: Thank you for the comment. We have added another section (“2.3. Optimization and validation of the extraction conditions”) where we explain the process of optimization, including the model used, the optimized response, and the validation of the model.

TPC seems to be used to abbreviate both “Total Phenolic Content” and “Total Phenolic Compounds”. Please use for a single abbreviation, I suggest “Total Phenolic Content” which is widely used in the bibliography together with Folin-Ciocalteu method.

RESPONSE: Thank you for the suggestion. The use of TPC was revised in order to abbreviate the expression “Total Phenolic Content” throughout the manuscript.

Legend Table 2: Some spaces are missing.

RESPONSE: Thank you for the comment. The names of phenolic compounds were corrected in the footnote of Table 2.

Some contradictory interpretation is also observed, for example:

Lines 122 and 123 “Besides, neither the extraction temperature nor the pH influenced the antioxidant capacity (P =0.2328 and 0.3635, respectively).” Later, in Lines 125-127: “ In our case, there was also observed an increasing trend (P> 0.05) in the antioxidant capacity when temperature was augmented, as well as when highly acidic and alkaline conditions were used.”

RESPONSE: Thank you for the comment. We only obtained a trend, since we have an ascending curve in the case of the temperature and a “U” curve in the case of pH, but they are not significant changes, as we explain with the expression “P>0.05”. However, for avoiding doubts about these results, we have explained it with a explicit sentence: “In our case, there was also an increasing trend (P> 0.05) in the antioxidant capacity when temperature was augmented, as well as when highly acidic and alkaline conditions were used. However, as we have mentioned before, these increases are not significant.”

References: Issue and pages information is missing in the references 16,17,18,19 and 21.

RESPONSE: Thank you for the comment. Properly corrections were performed on the references 16, 17, 18, 19, and 21 (now 18, 19, 20, 21, 22, and 24, respectively).

Round 2

Reviewer 2 Report

All the questions/corrections were answered. In my opinion the manuscript can be now published in Molecules Journal. The following minor revisions are requested:

Table 1: Please put as first the column of Run#;

Line 85: 7 "µM Trolox/g of dry matter" please replace by "µM TE/g of dry matter". Please indicate also if TPC is also in dry matter basis. you should uniformize all the units.